

# nCov2019: an R package for studying the COVID-19 coronavirus pandemic

Tianzhi Wu[1], Erqiang Hu[1], Xijin Ge[2] and Guangchuang Yu[1]

[1] Department of Bioinformatics, School of Basic Medical Sciences, Southern Medical University, Guangzhou, China
[2] Department of Mathematics and Statistics, South Dakota State University, Brookings, United States

## ABSTRACT

**Background:** The global spreading of the COVID-19 coronavirus is still a serious public health challenge. Although there are a large number of public resources that provide statistics data, tools for retrospective historical data and convenient visualization are still valuable. To provide convenient access to data and visualization on the pandemic we developed an R package, nCov2019 (https://github.com/YuLab-SMU/nCov2019).

**Methods:** We collect stable and reliable data of COVID-19 cases from multiple authoritative and up-to-date sources, and aggregate the most recent and historical data for each country or even province. Medical progress information, including global vaccine development and therapeutics candidates, were also collected and can be directly accessed in our package. The nCov2019 package provides an R language interfaces and designed functions for data operation and presentation, a set of interfaces to fetch data subset intuitively, visualization methods, and a dashboard with no extra coding requirement for data exploration and interactive analysis.

**Results:** As of January 14, 2021, the global health crisis is still serious. The number of confirmed cases worldwide has reached 91,268,983. Following the USA, India has reached 10 million confirmed cases. Multiple peaks are observed in many countries. Under the efforts of researchers, 51 vaccines and 54 drugs are under development and 14 of these vaccines are already in the pre-clinical phase.

**Discussion:** The nCov2019 package provides detailed statistics data, visualization functions and the Shiny web application, which allows researchers to keep abreast of the latest epidemic spread overview.

## INTRODUCTION

The COVID-19 pandemic emerged at the end of 2019 from Wuhan, China (*Zhu et al., 2020*; *Wang et al., 2020*). The virus has raged globally for more than 12 months and still in the process of accelerating its spread. Currently it remains an extremely serious public health challenge, affecting more than 220 countries worldwide. After a year-long battle, researchers have gained more insight into the transmission route, molecular structure,

Corresponding authors
Xijin Ge, Xijin.Ge@sdstate.edu
Guangchuang Yu,
gcyu1@smu.edu.cn

sequence information and pathogenesis of the novel COVID-19 virus (*Lippi et al., 2020*; *Ahn et al., 2020*; *Li et al., 2020*; *CNCB-NGDC Members and Partners, 2021*). Experts in epidemiology and data science also play a great role in this; they collected data and developed many convenient tools to deliver information on the latest infection numbers, high-risk areas and so on. The World Health Organization (*WHO, 2020*) and the national Center for Disease Control and Prevention (CDCs) publish authoritative data but usually on a daily basis update frequency, and the data often appear to lag behind. In contrast, statistics reported by aggregated news media, such as *DXY.cn (2021)*, *Worldometer (2021)*, *Our World in Data (n.d)* and *BNO News (2021)*, are usually updated more frequently and published before the CDCs. The Johns Hopkins Center for Systems Science and Engineering (CSSE), Baltimore, MD, USA integrates statistics from these data source and provides an online visual dashboard (*Dong, Du & Gardner, 2020*); it is the most popular and frequently updated platform we could find. Their efforts have facilitated the public to get the latest information on the spread of the epidemic.

Data analysis and customized visualizations are essential to the medical researchers and epidemiology experts, for analyzing changes of spreading, monitoring new outbreak trends, assessing current health measures and so on. Obtaining data is a prerequisite for data analysis. As time goes on, there are many websites and resources that provide COVID data. Currently, several R packages are available to provide different types of data. For instance, covid19.analytics (*Ponce & Sandhel, 2021*) provides virus sequence queries, as well as dashboard and SIR models; COVID19 (*Guidotti & Ardia, 2021*) and coronavirus (*Krispin & Byrnes, 2021*) packages provide detailed vaccine and case test data.Here, we provide an R package, nCov2019, which aims to access, visually explore and analyze epidemic related statistics data in R (Fig. 1). The nCov2019 package is more comprehensive in comparison to other R packages. It provides more data types including real-time and historical infection statistics, therapeutic and vaccine data. In addition, our tool provides several convenient and practical visualization functions and an interactive dashboard based on the Shiny web application with no code requirements in order to help users explore the data visually. The nCov2019 package was developed in Jan, 2020 and is one of the earliest R packages designed to query COVID data to support epidemiology modeling. It is available at a time when there were few data resources available and was cited in several academic articles (*Guo, Zhang & Zeng, 2020*; *Zhang et al., 2020*; *Zeng et al., 2020*; *Chagas et al., 2020*).

## MATERIALS & METHODS

The main purposes of developing nCov2019 are to reduce the barriers of data acquisition and to provide essential visualization functions, including dynamic visualization to monitor the spread of the virus in an easier and concise way. To achieve these goals, the nCov2019 package was designed with four main parts: data collection, data query and operation, geographic maps visualization and interactive dashboard.

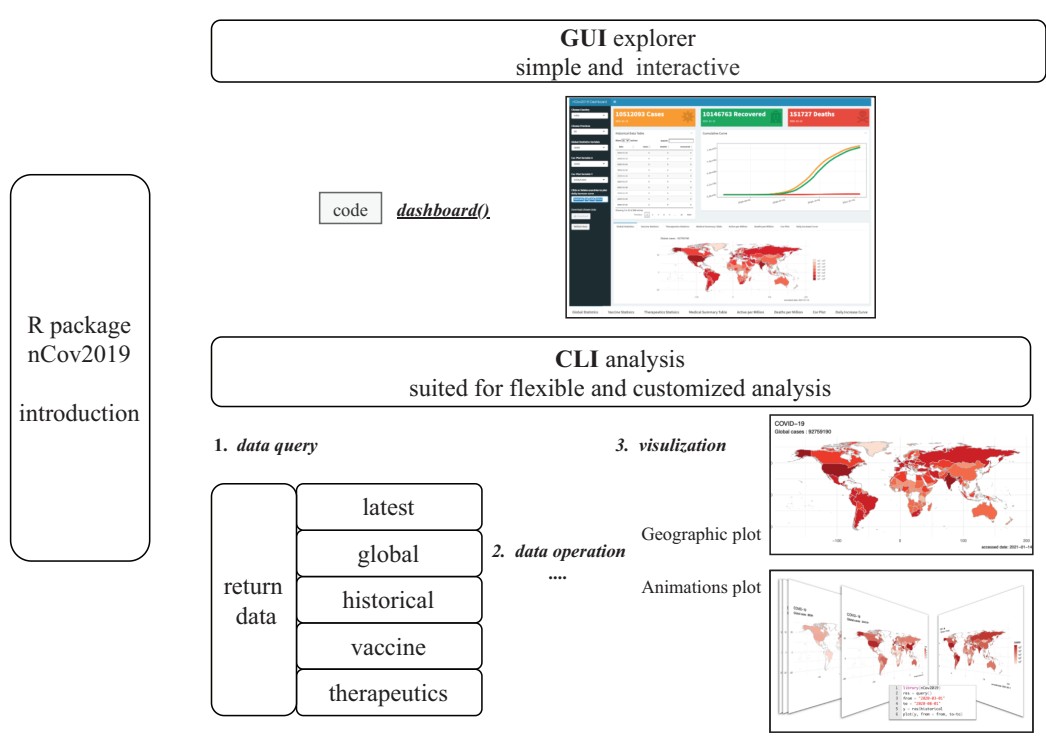

**Figure 1 The Diagram of nCov2019 architecture.** nCov2019 is designed for both command line (CLI part) and dashboard interaction analysis (GUI part), while *dashboard()* is the main entry for the GUI explore part, the *query()* is the main function used in CLI explore part. nCov2019 also provides a map visualization and an animation function, the latter can be used to draw dynamic maps.

## Data collection

The Statistics data usually contains the latest status and historical data. For the real-time latest status, we chose WorldoMeters as our data source, which has the high update frequency. Our historical data source is CSSE, which integrates data from multiple source centers and the data is reliable and timely. In addition, after 1 year of continuous battle of the COVID-19 virus, researchers have developed some vaccines, and at the same time, doctors have tried different treatment options. These data are more important than simple summary of confirmed, recovered and death cases. We also collected data on vaccine development and drug therapeutics progress from two website of raps.org (*RAPS, 2020a*; *RAPS, 2020b*) and these data can also be accessed directly in our nCov2019 package.

## Data query and operation

For ease of use, we wrapped the data fetching process into a simple function, *query()*. It usually needs to be executed only once in a session, then users can obtain five datasets. The latest data contains detailed statistics status at the time of query, and the historical data contains daily statistics for each country, which is often useful to model epidemic growth. We also provide a global summary to help understand the latest progress in the fight against the COVID-19 pandemic.
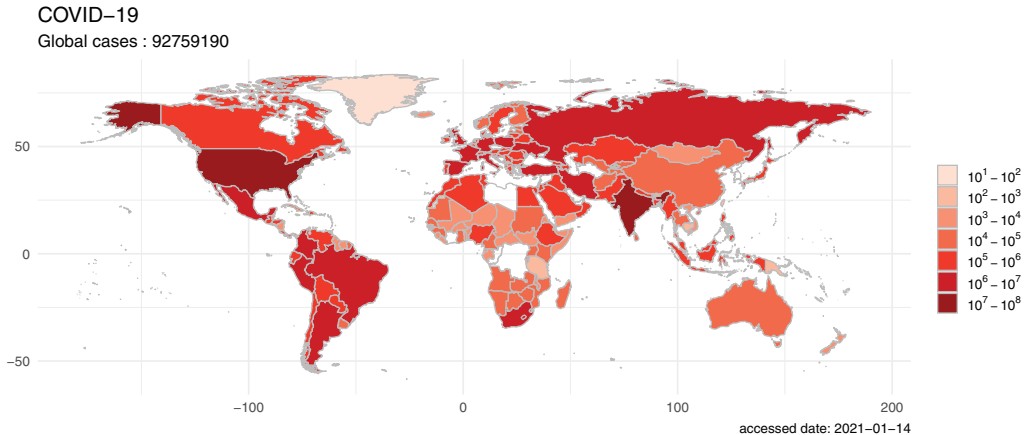

**Figure 2 The world map visualizes the global spread of the COVID-19.** The map was filled by colors indicating confirm cases of each country and the global view shows that the COVID-19 pandemic is a serious public health challenge. The plot was generated on 2021-01-14 with build in function of the nCov2019 package.

To facilitate downstream data analysis, we defined the '[' operator which mimics the API of data selection for data frame in R. So that data can be easily accessed by specific regions. For example, let *X* be the historical data, then *X[c("USA","UK")]* will return the historical data table for USA and UK only. These data are organized in long format and can be directly handed over to ggplot2 for plotting in R.

The vaccine and drug therapeutics development status can be queried at the same time. The data contains the latest information about candidate medication class, mechanism of vaccine, trade name for drugs and their current trial phase. Both datasets have the details information such as background, develop aims and trial details. These data are helpful for users to understand the medicine progress of epidemic prevention and control.

## Geographic maps

Geographic visualization is an effective way to observe the spatial patterns of virus spread. We provided built-in and convenient geographic map visualization functions within the nCov2019 package. The visualization functions were wrapped into a simple and easy-to-use command as *plot()*, so that users can plot the distribution of cases on the maps of the worldwide or national scope. For example, let *X* be the latest data in query result, then *plot(X)* will plot the world map contained global confirmed cases (Fig. 2).

To review and analyze the spread of COVID-19 epidemic situation, a more informative application is to draw dynamic geographic maps at multiple time points. Users can easily get the dynamic map by just specifying the start and end time in our tool. For instance, by performing *plot(X, from = "2020-03-01", to = "2020-08-01")* function, an animation will be generated to reflect transmission and spread dynamic of the COVID-19 outbreak during these time points (Fig. S1).
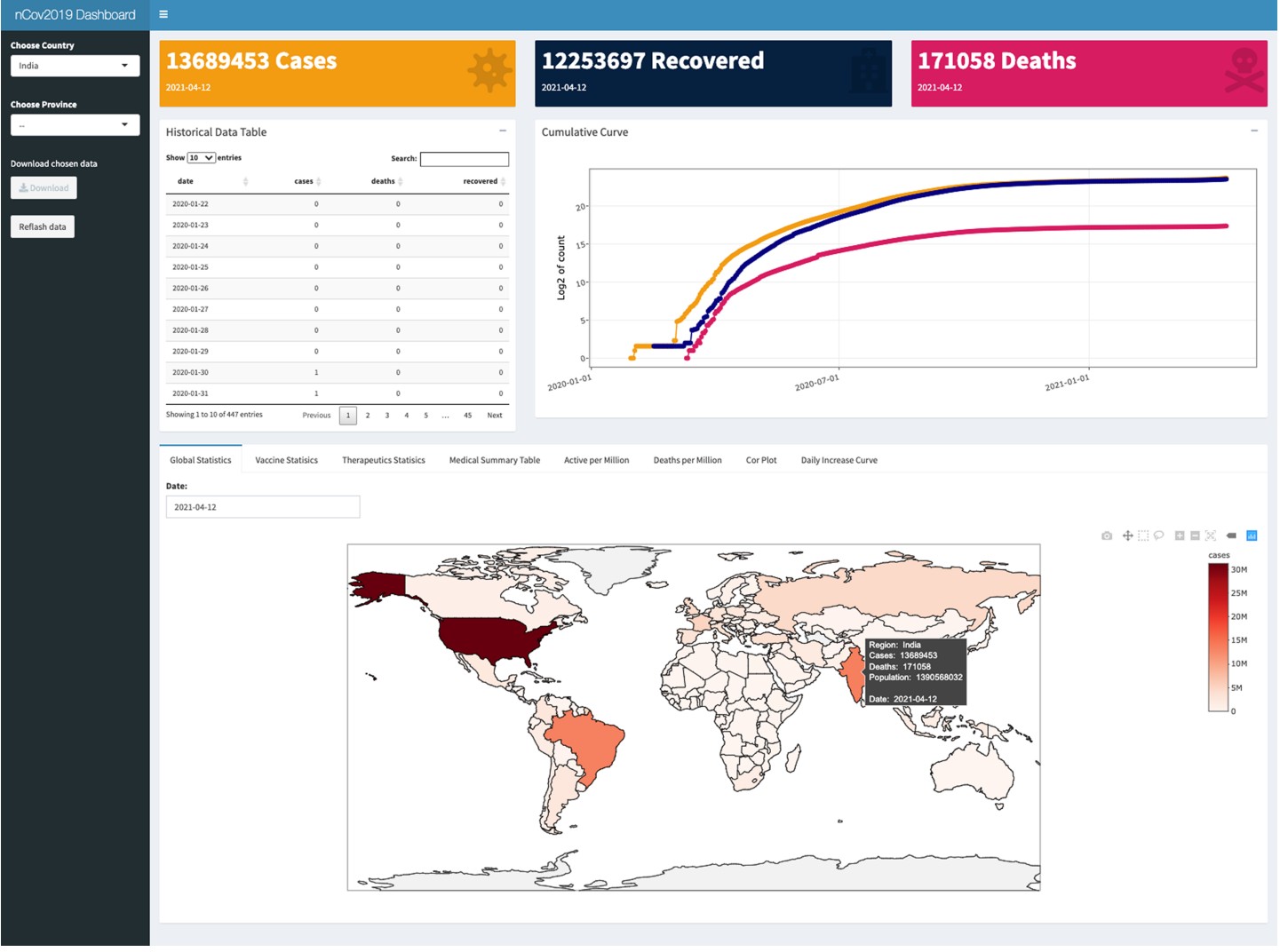

**Figure 3 The user interface of the dashboard provided in the nCOV2019 package.** The panel includes statistical data query, map visualization, interactive analysis and summary of the therapeutics and vaccine development. User could launch it with the function *dashboard()*.

## Interactive dashboard

We also developed an interactive web dashboard to help users to access and explore these datasets by interactive mouse clicks. Built with the RStudio Shiny framework, the dashboard could be launched with the function, *dashboard()*. It enables users to select their regions of interest and check both the historical and real-time data. The statistics of confirmed, deaths, and recovered cases will be displayed in the Dashboard header, followed by a downloadable statistics data table, nearby their cumulative curve (Fig. 3).

Multiple charts are designed on the bottom of dashboard. A chart of global statistics could display nine types of statistics on the world map, such as confirmed cases, active infected cases, number of detections, and population for each country and so on. Vaccine and therapeutics summary table is shown next to global statistics chart. For exploring the dataset easier, we provided an interaction plot; it can be used to explore the relationship
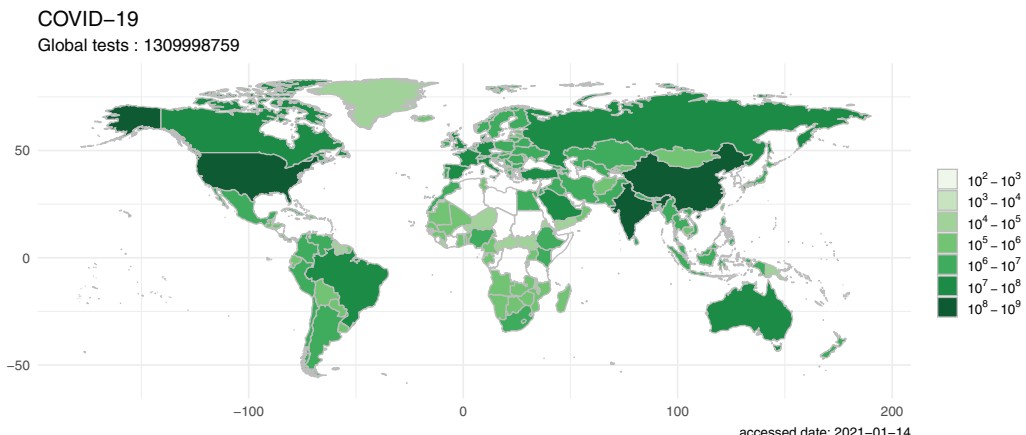

**Figure 4 The map visualized by nCov2019, reflecting the amount of COVID-19 testing in different countries around the world.** While Asia and North America performed the highest number of detection tests for the virus, Africa had the lowest number of detections, although this may be due to temporary lack or delay of reported data in Africa countries, it cause an alarm that more attentions should be given to African countries.

between any two of the twenty different statistics, such as whether there is an association between confirmed cases number and the total detection number. Finally, we designed a curve plot to reflecting the growth intensity by daily increase cases for each country. This chart could be used to monitor the outbreak strength over time. One of its practical applications is to determine whether a second wave of outbreaks is occurring.

## RESULTS

At present, the global epidemic crisis is still serious. According to data calculations from the nCov2019 package, as of January 14, 2021, the number of worldwide confirmed cases is 91,268,983 (Fig. 1), the cumulative number of deaths has reached to 1,951,790. Two country have more than 10 million confirmed cases (23,616,515 confirmed cases, 236,631 per million population in USA and 1,051,2831 confirmed cases, 7,578 per million population in India), which indicating that there is still great pressure for COVID-19 prevention and control.

Detection of infected persons and cutting off the transmission route are the key solutions to curb the virus spread. Detection status could be visualized by using the visual map function provided in our package (Fig. 4). Intuitively, it can be seen that Asia and North America performed the highest number of detection tests for the virus, compared to Africa, which had the lowest number of detections. This may be due to a temporary lack or delay of reported data in Africa countries. But given the situation of general economic backwardness and large population, it is needed for other countries to support the fighting against the pandemic in Africa.

Over the past year, the epidemic has shown multiple peaks in many countries, such as Australia, Japan, Italy, Germany and China. The epidemic in some countries were under control by middle in 2020, but then the confirm cases has rebounded (Fig. S2). While these countries are now at the tail end of the second wave of the epidemic, the

possibility of a third wave in the future is a cause for alarm. Some countries, such as USA, India, Russia, and South Africa, are still in the midst of a severe and rapid increase.

Researchers are playing an active role in the fight against the virus. There are currently 51 vaccines in development or in use worldwide. Due to the emergencies, most vaccines and regimens are using a simultaneous multi-clinical trial approach. Fourteen of these vaccines are already in pre-clinical phase. As for the therapeutics, there are currently 54 candidates, including HIV protease inhibitor, IL-6 receptor agonist, HIV-1 Rev protein inhibitor, Autologous adipose-derived stem cells, and Monoclonal antibody.

All of the above information is available directly in the R package either using command line or dashboard (Fig. 3). Detailed vignette including the usages of data acquisition and visualization functions could be found in the Supplemental File and the package is hosted on CRAN (https://cran.r-project.org/package=nCov2019).

## DISCUSSION

We provide practical and concise tools for searching outbreak data, as well as vaccine and treatment-related data. The data are sourced from reliable platforms. As a highlight, our nCov2019 package is designed not only to help clinicians and other researchers to obtain statistics table, but also to help them reducing the difficulty of visualizing comprehensive data. Researchers could utilize our package to track the latest statistic status, conduct retrospective analysis or establish predictive models. Our package was collected in several resource list, such as the COVID-19 Coronavirus Disease Resources (*COVIRUSD, 2021*) and Top 100 R resources on Novel COVID-19 Coronavirus (*Stats and R, 2020*), and was served as the back end for the COVID-19 Pandemic Analysis Platform (*ACM Digital Library, 2021*).

The epidemic has been going on for over a year and there is now a lot of online dashboard (*Alenezi, 2021*) or websites (*WHO, 2021*; *Hahn, 2021*) with great appearance design. For the public who just want to know the latest status of the epidemic, it is the most convenient way by simply browse from the internet. In contrast, our R package is designed for both command line and dashboard interaction analysis, it is more suitable for researchers and data analyzers who wish to explore the epidemic data by themselves. At present, our package has provided data support for several academic research (*Guo, Zhang & Zeng, 2020*; *Zeng et al., 2020*; *Chagas et al., 2020*; *Ran et al., 2020*; *Zhuang et al., 2020*), and we hope our tool will continue to facilitate similar studies.

## CONCLUSIONS

Our nCov2019 package reduces the barrier for researchers and public health officials in obtaining comprehensive, up-to-date data on this ongoing outbreak. With this package, epidemiologists and other scientists can directly access data, produce convenient and rich visualization images, facilitating mathematical modeling and forecasting of the COVID-19 outbreak. The interactive web dashboard is accessible to the general public and could also be easily customized by researchers to produce other dashboards or track other countries. We hope our tool will be helpful to the COVID-19 relevant research work.

### Funding

This work was supported by the startup fund from Southern Medical University. The funders had no role in study design, data collection and analysis, decision to publish, or preparation of the manuscript.

### Grant Disclosures

The following grant information was disclosed by the authors:
Startup fund from Southern Medical University: G618289088.

### Competing Interests

The authors declare that they have no competing interests.

### Author Contributions

- Tianzhi Wu conceived and designed the experiments, performed the experiments, analyzed the data, prepared figures and/or tables, authored or reviewed drafts of the paper, and approved the final draft.
- Erqiang Hu performed the experiments, analyzed the data, prepared figures and/or tables, and approved the final draft.
- Xijin Ge contributed to the dashboard, authored or reviewed drafts of the paper, and approved the final draft.
- Guangchuang Yu conceived and designed the experiments, performed the experiments, analyzed the data, prepared figures and/or tables, authored or reviewed drafts of the paper, and approved the final draft.

### Data Availability

The package is available on CRAN:
https://cran.r-project.org/package=nCov2019.

### Supplemental Information

Supplemental information for this article can be found online at http://dx.doi.org/10.7717/peerj.11421#supplemental-information.

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
