# Peer review of "nCov2019: an R package for studying the COVID-19 coronavirus pandemic"

_PeerJ, doi:10.7717/peerj.11421_

## Round 0.1 · original submission · Major Revisions

Our reviewers have several important suggestions for the improvement of your tool and manuscript. Please address all of them thoroughly. Regarding reviewer 3's comments on novelty, I must note that novelty is not a requirement for publication in PeerJ, but that we do require that every manuscript be a clear addition to the literature. You should therefore describe the currently-available tools and explain how your tool differs from /improves upon them.

·

Basic reporting

No comment

Experimental design

No comment

Validity of the findings

No comment

Additional comments

Overall, the flow of the paper is easy to follow and the work is very meaningful. I have a few specific comments:

Title: "An analytics tool ..." The wording "analytics tool" is a bit unclear. After reading the paper, my understanding is that nCov2019 is an R package that provides historical covid infection data and visualization. It also provides some forecast utility. I suggest the authors consider making the title more precise.

Abstract Line 30: "Although there are a large number of public tools that provide statistics data" (Line 46): "Although there are many tools and web services
provide epidemic data, there are still fewer tools provide historical data for retrospective
analysis"
-> I wonder if authors can provide more context on what data is available/easily accesible before this work and what is not available/easily accessible. Some references will be extremely helpful. By providing more context of the existing public tools, the contribution of this paper will be clearer and more appreciated. Authors should provide more details about what is missing in the current tools such as refs 3,10.

Reviewer 2 ·

Basic reporting

The manuscript by Tianzhi Wu et al “nCov2019: An analytics tool for studying the COVID-19 coronavirus pandemic” present nCov2019 an R package and a Shiny application for data visualization and analysis to follow the SARS-CoV-2 epidemic at country-level. Data analysis and visualization is an essential tool for exploring and communicating findings in medical research. I find the shiny web application intuitive and responsive.

Major comments:

My main concern is related to method and discussion section:

In lines 139 to 144, the authors explain which data and statistics are included in the app but no definitions, details or formula are provided neither epidemiological jargon is used. For example, according to Rothman et al [Rothman K, Greenland S. Modern epidemiology. Philadelphia, PA: Lippincott-Raven Publishers; 1998.] the epidemiological term for “Mortality rate was calculated by dividing total deaths by total confirmed case” should be the case fatality rate. Please, add a definition for each term, specify which graphs are provided, and use epidemiologic terms.

In lines 145 to 150, the authors explain the forecast module. Statistical details on this module are scarce and mixed with results. Please, specify statistical details about the model and parameters used for forecasting. If authors have performed validation on models with data from Italy, the validation method should be explained in the Methods section and the results of this validation in the Results section. The use of data only from Italy for validation should be justified. Moreover, verify citation format and think if it's redundant to cite the paper and the web page of the paper's author.

In figure 2 the nCOV2019 dashboard is presented. However, the image does not coincide with the linked app in the manuscript (http://www.bcloud.org/e/). Please, verify and correct.

Author’s vignette (https://guangchuangyu.github.io/nCov2019/) on nCov2019 package is full of interesting details and good examples. The authors could include the vignette as supplementary material.

On line 72, 114 and 180, the authors defend that they provide data to establish predictive models. In this reviewer's opinion, this statement should be lowered. One thing we have learned after 9 months is the complexity of this epidemic. At present, we can estimate incidence trends but we are far from having reliable predicting models. Forecasting of a pandemic can only be done based on a wide variety of parameters such as transmission rate, incubation period, the number of carries (symptomatic or asymptomatic), quarantine impact, and many other environmental and social factors.

The discussion should be expanded in terms of the strengths and limitations of the provided analytic tool. Moreover, the authors mention some examples of worldwide applications of the COVID-19 epidemic but do not provide a comparison. It would be valuable to show the differences between them to provide a comparison and emphasize the main contribution of the present work.

Figure 1, 2 and 3: Figures are blurred. Please provide a better quality of the graph.

Experimental design

no comment

Validity of the findings

no comment

Additional comments

Minor comments:

Introduction
On lines, 55 to 57 authors presented a list of top countries in relation to the number of confirmed cases. A rate, cases divided by population, would result in a fairer comparison. Otherwise, you are only reporting the number of cases of the highest populated countries.

In line 63, CDC is mentioned for the first time. Please, provide what the CDC acronym stands for.

In line 64, the authors mention “and most of them are presented using online dashboards”. Please, could you cite and explain the most relevant examples of these online dashboards from WHO or from an specific country?

On lines 66 to 70, in relation to what is convenient to analyze of this epidemic, the authors said: “Thus, more specific historical data is needed. However these data are not well obtained at present, as many data sources only provide real-time aggregated data. There is still a lack of clean, easily accessible historical data source and analysis tool”. I agree with this point. However, I miss a proposal on how data should be. Please, could you provide, in your point of view, how should this data be? You can focus your explanation on a specific country or in a specific type of epidemiological analysis.
Results
On line 153 “According to data calculations from the nCov2019 package, as of August 10th,” and in Figure 3 data is from October 10th. Please clarify.

On line 161 “In these 43 countries, case 161 dead rate ranges from 0.17% (219/127778) in Qatar to 10.34% (36140/349494) in Italy”. According to Rothman et al the epidemiological term should be case fatality rate.

On line 164 “By exploring the latest growth rate, we found that although the United States, India and 165 Brazil have received public attention due to the huge total number of confirmed cases”. A date is necessary to contextualize. In a few weeks, this statement could false. Moreover, authors should explain how they have estimated the latest growth rate. The explanatory sentence on line 144 on the methods section seems to cut off.

Discussion
On line 176 “There are currently lots of data sources 3,10”. References are about online interactive dashboards one from WHO and another from Johns Hopkins University. There are many open data sources, most listed on this web page from the National Institute of Health –United States (https://datascience.nih.gov/covid-19-open-access-resources). And several online interactive dashboards such as those cited or: https://covid19.healthdata.org/, https://ubidi.shinyapps.io/covid19world/.

On line 184 “Our package was collected in several 185 resource list and served as backend for several plotforms”. Please, correct typo on platforms. Moreover, reference 11 on “Coronavirus Tracker. https://waiter.john-coene.com” is not about an online interactive dashboard. Please, correct the link or delete the reference.

·

Basic reporting

The article « nCov2019: An analytics tool for studying the COVID-19 coronavirus pandemic » is an article to present the R package nCov2019. The package intends to provide a way of accessing historical COVID19 data directly from R and to provide analysis tools in the form of a shiny application and plot functions. The package is available through github and works, the features presented can be reproduced.
The literature is succinct, and almost no reference to other dataset or dashboard is provided (Apart from WHO and John Hopkins).
The authors mention the blog post https://www.statsandr.com/blog/top-r-resources-on-covid-19-coronavirus/ which refers to 39 shiny app about COVID and 8 other R packages about COVID19 data. The authors should at least refer to them and comment on the difference.

Experimental design

no comment

Validity of the findings

While I appreciate the effort and the work done, I have several strong critics, that I would summarize in the following points:

1. The main point is that I do not see the novelty of what is proposed here. Several historical datasets are easily available in one command line in R, and plenty of advanced visualization tools are now available, which are not cited in the manuscript. The authors only cite the John Hopkins dataset, but do not compare their tools and dataset with this one, nor do they clarify the novelty of the proposed tool.

2. The grammar of the R package is not clear. The package is not published on the CRAN, and the documentation to succinct to help

3. There are multiple problems with the vizualisation tools, especially with the shiny app (wrong scales, missing interactivity on maps, missing axis legend, errors on data selection etc). The local app and the online ones are not the same at all. The online app is extremely slow. The overall impression is that the app is not finished nor properly checked.

4. Some indicators are strongly problematic, namely the growth rates, which are calculated on a daily basis when it is known that the cases fluctuate strongly on a daily basis. Some other are undefined and unknown, like the health rate. The prediction given by forecast are rather uninformative, and not based on any epidemiological background.

5. The data number are different from other institutional COVID19 dataset, and I do not see any comments about it, nor validation or comparison of the present dataset with other institutional dataset.

The first four points are critical for me. Points 2, 3 and 4 would require as subsequent work, as the overall impression is that the package is not finished and rather in a “work in progress” stage. The first point is essential and from what is presented here, I do not think that the package bring something new concerning visualization or data about COVID19. I would therefore not recommend publication in peerJ, and advise the authors to target a R specific publication when the package is finished and made available on CRAN

The detail of each point is given in the attached pdf

---

## Round 0.2 · Major Revisions

Reviewer #3 feels that you may have missed the detailed points they had highlighted in the PDF they had attached to their review. Please address all of their points (both in that PDF and in the current review)

·

Basic reporting

The article « nCov2019: An analytics tool for studying the COVID-19 coronavirus pandemic » is an article to present the R package nCov2019. The package intends to provide a way of accessing historical COVID19 data directly from R and to provide analysis tools in the form of a shiny application and plot functions. The authors made substantial changes during their first review, improving both their manuscript and package, and include now important new features, such as vaccine list, or the possibility to generate animation from map plots. Despite these improvements, there are still majors problems in the author's article and package: the package is still not on CRAN, thus unable to guaranty interoperability across platforms and changes continuity; the comparison with other R package (submitted on CRAN) providing similar tools and data is not done; there is no mention of ourworldindata data explorer and dataset in intro or discussion; the access of province historical data is not integrated in the package syntax anymore; the package documentation is too succint or absent for most of the R functions; the dashboard has still multiple flaws. I therefore ask for major revisions.
It seems that the authors did not read the associated pdf of my previous review, which contained the detail of each of my points with associated print screen. I will this time put all the detail in the present review.

Experimental design

- I understand that the authors feel that a github repository is enough, but I still insist that a CRAN submission should be required before the acceptance of the current paper. The authors actually gave an example of this necessity, by moving their package from https://github.com/GuangchuangYu/nCov2019 to https://github.com/YuLab-SMU/nCov2019. Here, users that want to update the package will not get the latest version. Furthermore, CRAN is a guaranty that the package works on all platforms, respects some standards about documentation and vignettes, and has a proper documentation of changes. But I agree that, as for any CRAN package, a github repository is needed for issues handling and development. The submission to CRAN only take few days.

Validity of the findings

- My first point was already mention in my previous review (attached pdf): the authors do not mention ourworldindata.org, which is one of the main data source in the world about COVID19 pandemic (with CSSE), with a complete data explorer containing an important amount of variable selection, maps, possibility to compare countries and to select dates, possibility to have animations etc: https://ourworldindata.org/coronavirus-data-explorer. Furthermore, they have a single complete dataset of historical data in long format of all their variable accessible on a fixed url, updated daily, and so accessible directly in R in a single command line:
"x <- read.csv("https://covid.ourworldindata.org/data/owid-covid-data.csv") "
Which is something equivalent to
"library(nCov2019)
res <- query()
x <- res$historical" .
This should be discussed, and the author's sentence "However, to accessed complete data from online source, users need to crawl, parse and store info from their web pages. Additional manual check, data conversion and data clean are required before the data can be used in analysis or model fitting" in the author manuscript should be removed, and the novelty description of the package should be updated.

- The authors did not discuss in intro the difference between their work and existing R package on the same subject already published in CRAN (some with associated peer reviewed paper). Among others: https://cran.r-project.org/web/packages/COVID19/index.html ; https://cran.r-project.org/web/packages/coronavirus/index.html ; https://cran.r-project.org/web/packages/covid19.analytics/index.html

I will focus the rest of my review on the tool presented, as it is the core of the article. Here are the majors points:

- Although the syntax of the package is easier to understand now, the ability to get the province from the dataset using the "[]" syntax described in the manuscript disappeared. The user has to guess and find that "res$historical$province" provide a data.frame with province data (this is not described in vignette), on which "x["beijing"]" does not work for example. Furthermore, the list of countries with province changed: I get
"Australia" "Canada" "China" "Denmark" "France" "Netherlands", but can't see Italia nor US.

- The documentation entry for "query" has virtually no information: the output is not described. More generally, the documentation of the package is globally to succinct (no example in doc entries, no description of the output of the function, etc)

- the dashboard has several problems. As the added value that the authors want to give to their work is the possibility to inspect the data, several improvements and corrections are needed:
1) The maps are not interactive. As suggested in my previous review, the authors should at least use leaflet in their shinyapps, so that the user can zoom in and out and get information when hovering the countries.
2) there is not date selection on the dashboard for the map
3) The files to download are with .tsv extension instead of .csv
4) The fact that the death, active case and positive case cumulated are on the same scale does not allow to see the death trend properly.
5)The activeperillion and deathpermillion plots have no x axis, and the y axis should be understandable text instead of variable name
6) the correlation plot uses a single color when different color would help to distinguish between countries. Furthermore, the size of the points changes, but the scale of the point size is not indicated
7) The plot with the daily changes indicates in the plotly legend log(diff+1) = 6, without explanation. The y scale is not comprehensible in the present state.

---

## Round 0.3 · accepted · Accept

I am glad to accept your manuscript for publication in PeerJ.

·

Basic reporting

The authors have corrected the manuscript properly.

Experimental design

ok

Validity of the findings

ok

Additional comments

Thank you for the efforts, and for submitting to the CRAN.